# Multimodal AutoML on Tables with Text Fields

**Xingjian Shi**[*]        `xjshi@amazon.com`
**Jonas Mueller**[*]     `jonasmue@amazon.com`
**Nick Erickson**       `neerick@amazon.com`
**Mu Li**               `mli@amazon.com`
**Alexander J. Smola**     `alex@smola.org`

Amazon Web Services

## Abstract

We consider the design of automated supervised learning systems for data tables that not only contain numeric/categorical columns, but text fields as well. Here we assemble 15 multimodal data tables that each contain some text fields and stem from a real business application. Over this benchmark, we evaluate numerous multimodal AutoML strategies, including standard two-stage approaches where NLP is used to featurize the text such that AutoML for tabular data can then be applied. We identify practically superior strategies based on multimodal adaptations of Transformer networks and stack ensembling of these networks with classical tabular models. Compared with human data science teams, the best fully automated methodology[2] discovered through our benchmark manages to rank 1st place when fit to the raw text/tabular data in two MachineHack prediction competitions and 2nd place (out of 2380 teams) in Kaggle's Mercari Price Suggestion Challenge.

## 1 Introduction

Despite recent data proliferation, the practical value of machine learning (ML) remains hampered by an inability to quickly translate raw data into accurate predictions. Automatic Machine Learning (AutoML) aims to address this via pipelines that can ingest raw data, train models, and output accurate predictions, all without human intervention [35]. Given their immense potential, many AutoML systems exist for data structured in tables, which are ubiquitous across science/industry [25, 30, 58].

Many data tables contain not only numeric and categorical fields (together referred to as *tabular* here), but also fields with free-form text. For example, Table 1 depicts actual data from the website Kickstarter. These contain multiple text fields such as the title and description of each funding proposal, numerical fields like the goal amount of funding and when the proposal was created, as well as categorical fields like the funding currency or country. This paper considers tables of this form where rows contain IID training examples (each with a single numeric/categorical value to predict, i.e. regression/classification) and the columns used as predictive *features* can contain text, numeric, or categorical values. We refer to the value in a particular row and column as a *field*, where a single text field may actually contain a long text passage (e.g. a multi-paragraph item description). Despite their potential commercial value, there are currently few (automated) solutions for machine learning with this sort of data that jointly contain numeric/categorical and text features, which we refer to as *multimodal* or *text/tabular* data. Applying existing AutoML tools to such data thus requires either manually featurizing text fields into tabular format [5, 29], or ignoring the text. Alternatively, one can use existing natural language processing (NLP) tools to model primarily just the text [11, 27, 28, 34, 52].

---

[*]Equal contribution.

[2]Available to easily run on your own data through: `https://github.com/awslabs/autogluon`

Submitted to the 35th Conference on Neural Information Processing Systems (NeurIPS 2021) Track on Datasets and Benchmarks. Do not distribute.

This paper considers design choices for automated supervised learning with multimodal datasets that jointly contain text, numeric, and categorical features. Even though text commonly appears along with numeric/categorical fields in enterprise data tables, how to automatically analyze such multimodal data has not been well studied in the literature. This stems from a lack of published benchmarks, as well as existing beliefs that basic featurization of the text [14, 29] should suffice for tabular models to exhibit strong performance. Here we introduce a new benchmark of 15 multimodal text/tabular datasets from real business applications (Section 3), and provide the first comprehensive evaluation of generic strategies for supervised learning with such data (Section 7). In particular, we consider: multimodal neural networks that jointly operate on text and tabular inputs (Section 4), featurizing text for tabular models (Section 5), as well as ensemble combinations of text (or multimodal) neural networks and tabular models (Section 6).

Note that we write *AutoML* to describe any modeling strategy that is robustly performant across a diverse set of datasets without manual adjustments. The AutoML method promoted in this paper (stack ensembling of tabular models with a multimodal Transformer network) is simply the strategy that happened to perform best in our systematic analysis of various modeling strategies over the proposed benchmark. Among other discoveries, our benchmark reveals that the conventional strategy of neural embeddings to featurize text for tabular models is suboptimal. We hope the public benchmark and open-source tooling introduced here spurs further research in this important practical direction.

## 2  Related Work

Today, tools for automated learning with text data remain scarce (e.g. this dearth forced Blohm et al. [5] to turn to tabular AutoML tools for automated text prediction). Instead modern NLP applications primarily require experts who unanimously favor Transformer networks as their model of choice for text [13, 50, 52]. However existing methods to input numeric/categorical features into Transformers remain rudimentary [52] and fail to outperform the best tree models for tabular prediction [33]. While seemingly relevant, recent work on Transformers for understanding structured text tables [12, 69] addresses different tasks than the multimodal text/tabular supervised learning studied in this paper.

The use of tabular models together with Transformer-like text architectures has received limited attention [39, 63], and it remains unclear how to optimally leverage their complementary strengths for multimodal data (due to lack of benchmarks). In contrast, a number of entirely-neural architectures have been proposed for multimodal settings [36, 53, 54, 66]. However the vast majority of these are for {image, text} data [2, 51, 55, 56], but the gap between neural networks and alternative models is far greater for images than for tabular data [33].

Large, sufficiently diverse/representative, public benchmarks have spurred significant progress in tabular AutoML [15, 16, 25, 71] and NLP [23, 41, 48, 64]. However we are not aware of any analogous benchmarks for evaluating multimodal text/tabular ML. There do exist a few miscellaneous text/tabular datasets scattered throughout popular ML data repositories [1, 61], but these are mostly small academic datasets that are not representative of modern applications with significant practical value. In contrast, multiple prediction competitions each involving a single real-world text/tabular dataset have been held, but winning solutions have heavily relied on dataset/domain-specific tricks [46]. Here we aggregate multimodal datasets from competitions and other industry sources into one benchmark that aims to reveal unifying principles for powerful generic modeling of this form of data.

| name | desc | goal | country | currency | created_at | final_status |
|---|---|---|---|---|---|---|
| The Secret Order - The Game that gives back Gl... | Can you trust your friends? Solve the puzzle? ... | 5000.0 | GB | GBP | 1424101105 | 0 |
| Booker Family Foods. Home made, the way food s... | Community based, home-made-foods producer, to ... | 2500.0 | US | USD | 1404617242 | 0 |
| J.A.E.S.A : Next Generation Artificial Intelli... | A true next generation AI with the ability to ... | 30000.0 | CA | CAD | 1399078600 | 1 |

Table 1: Example of data in our multimodal benchmark with text (*name*, *desc*), numeric (*goal*, *created_at*), and categorical (*country*, *currency*) columns. From these features, we want to predict if a Kickstarter project will reach its funding goal or not (*final_status*).

## 3  Benchmarking Multimodal Text/Tabular AutoML

We aim to design practical systems for real-world data tables that often contain text. The empirical performance of our design decisions is thus what ultimately matters. Representative benchmarks comprised of many diverse datasets are critical for proper evaluation of AutoML, whose aim is to reliably produce reasonable accuracy on arbitrary datasets without manual user-tweaking. Thus we introduce the first public benchmark for evaluating multimodal text/tabular ML, which is comprised of 15 tabular datasets, each containing at least one text field in addition to numeric/categorical columns. Our new benchmark is publicly available, as is the code to reproduce all results presented in this work (and also to recreate our modified benchmark datasets from the original data sources).

Our benchmark strives to represent the types of ML tasks that commonly arise in industry today. Appendix B provides detailed descriptions of each dataset. In creating the benchmark, we aimed to include a mix of classification vs. regression tasks and datasets from real applications (as opposed to toy academic settings) that contain a rich mix of text, numeric, and categorical columns. Table 2 shows it is comprised of datasets that are quite diverse in terms of: sample-size, problem types, number of features, and type of features. 11 of the datasets contain more than one text field (with 28 text fields in the airbnb dataset). These text fields greatly vary in the amount of text they contain (e.g. short product names vs. lengthy product descriptions/reviews). The data (and text vocabulary) stem from a mix of of real-world domains spanning: e-commerce, news, social media, question-answering, and product listings (jobs, projects, films, Airbnb). Subsequent accuracy results from Table 3 indicate the 15 underlying prediction problems also vary greatly in terms of both difficulty and how the predictive signal is divided between text/tabular modalities. To reflect real-world ML issues, we processed the data minimally (beyond ensuring the features/labels correspond to meaningful prediction tasks without duplicate examples) and thus there are arbitrarily-formatted strings and missing values all throughout. Systems that can perform well across the diverse set of 15 benchmark datasets are thus likely to provide real-world value for an important class of applications.

Each dataset in our benchmark is provided with a prespecified training/test split (usually 20% of the original data reserved for test set). Methods are not allowed to access the test set during training, and for validation (model-selection, hyperparameter-tuning, etc.) instead must themselves hold-out some data from the provided training data. As the choice of training/validation split is a key design decision in AutoML, we leave this flexible for different systems to choose in the learning process. To facilitate comparison between the novel AutoML strategies presented in this paper, we always used the same AutoGluon-provided training/validation split, which is stratified based on labels in classification tasks. Our use of other AutoML frameworks beyond AutoGluon (e.g. H2O) allows each framework to choose their own data splitting scheme.

## 4  End-to-end Multimodal Learning with Text/Tabular Neural Networks

We now outline the many possibilities that must be considered in AutoML for multimodal data tables with text. Key design choices include what models to use (and for which features), and how to optimally combine different models within an overall ML pipeline. Using our benchmark, we

| Dataset ID | #Train | #Test | #Cat. | #Num. | #Text | Task | Metric | Prediction Target |
|---|---|---|---|---|---|---|---|---|
| prod | 5,091 | 1,273 | 1 | 0 | 1 | multiclass | accuracy | sentiment associated with product review |
| airbnb | 18,316 | 4,579 | 37 | 24 | 28 | multiclass | accuracy | price of Airbnb listing |
| channel | 20,284 | 5,071 | 1 | 15 | 1 | multiclass | accuracy | news category to which article belongs |
| wine | 84,123 | 21,031 | 0 | 2 | 3 | multiclass | accuracy | which variety of wine |
| imdb | 800 | 200 | 0 | 7 | 4 | binary | roc-auc | whether film is a drama |
| jigsaw | 100,000 | 25,000 | 2 | 27 | 1 | binary | roc-auc | whether social media comments are toxic |
| fake | 12,725 | 3,182 | 2 | 0 | 3 | binary | roc-auc | whether job postings are fake |
| kick | 86,502 | 21,626 | 3 | 3 | 3 | binary | roc-auc | whether proposed Kickstarter project will achieve funding goal |
| ae | 22,662 | 5,666 | 3 | 2 | 6 | regression | $R^2$ | price of American-Eagle inner-wear items on their website |
| qaa | 4,863 | 1,216 | 1 | 0 | 3 | regression | $R^2$ | subjective type of answer (in relation to question) |
| qaq | 4,863 | 1,216 | 1 | 0 | 3 | regression | $R^2$ | subjective type of question (in relation to answer) |
| cloth | 18,788 | 4,698 | 2 | 1 | 3 | regression | $R^2$ | customer review score for clothing item |
| mercari | 100,000 | 25,000 | 3 | 0 | 6 | regression | $R^2$ | price of Mercari online marketplace products |
| jc | 10,860 | 2,715 | 0 | 2 | 3 | regression | $R^2$ | price of JC Penney products on their website |
| pop | 24,007 | 6,002 | 1 | 2 | 1 | regression | $R^2$ | online popularity of news article |

Table 2: The 15 multimodal datasets that comprise our benchmark. '#Cat.', '#Num.' and '#Text' count the number of categorical, numeric, and text features in each dataset, and '#Train' (or '#Test') count the training (or test) examples. In PDF, click on each Dataset ID for link to original data source.

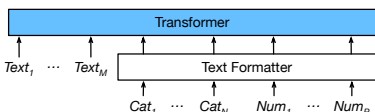

(a) *All-Text*. Convert numeric and categorical values into additional text tokens.

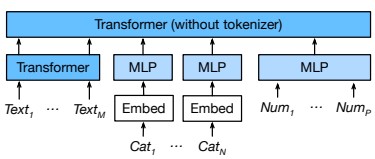

(b) *Fuse-Early*. Transformer operates on learned embeddings for each feature.

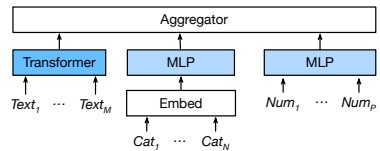

(c) *Fuse-Late*. Separate branches encode each modality, and aggregated via mean/max/concat.

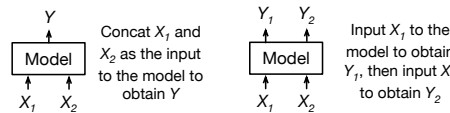

(d) Notation used in these figures.

Figure 1: Options for fusing modalities in *Multimodal-Net* (Section 4.2). Two dense layers (not shown) are added on top of each network in (a)-(c) to output a prediction (real value for regression, logit vector for classification). Over our benchmark, option (c) aggregated with concatenation performs best and is the chosen *Multimodal-Net* architecture in our proposed AutoML strategy.

present a systematic study that aims to cover the major variants of modeling paradigms used by practitioners today, including: NLP models to featurize text for tabular models [5, 14, 29], ensembling of independently-trained text and tabular models [46], or end-to-end learning with neural networks that jointly operate on inputs across text and tabular modalities [36, 52, 53]. In this section, we first consider the latter paradigm of multimodal neural network models, which in subsequent sections are also considered for text featurization and ensembling with tabular models.

## 4.1 Transformer Models for Text

We first consider solely inputting the text into our neural network and then discuss how to extend the network to additional numeric/categorical inputs in Section 4.2. While many neural architectures have been proposed to model text, pretrained Transformer networks now dominate modern NLP. These models are first pretrained in an unsupervised manner on a massive text corpus before being fine-tuned over our (smaller) labeled dataset of interest [13, 52]. This allows our supervised learning to benefit from information gleaned from the external text corpus that would otherwise not be available in our limited labeled data. The Transformer also effectively aggregates information from various aspects of a training example, using a *self-attention* mechanism to contextualize its intermediate representations based on particularly informative features [62]. Since BERT [13] first demonstrated the power of Transformer pretraining via Masked Language Modeling (MLM), superior pretraining techniques have been developed. RoBERTa [45] dynamically generates masks and pretrains on a larger corpus for a longer time, employing the same MLM objective as BERT in which random tokens are masked for the Transformer to guess their original value. ELECTRA [10] is an alternative pretraining technique in which a simple generative model randomly replaces tokens and the Transformer must classify which tokens were replaced.

Given a dataset with multiple text columns, we feed the tokenized text from all columns jointly into our Transformer (with special [SEP] delimiter tokens between fields and a [CLS] prefix token appended at the start [13]), as detailed in Appendix A.2. A single embedding vector for all text fields is obtained from the Transformer's representation at the [CLS] position after feeding the merged input into the network [13]. Similarly, just a single text field can be embedded via the Transformer's vector representation at the [CLS] position, after feeding only this field into the network.

## 4.2 Extending Transformer Architectures to Multimodal Inputs

In many multimodal datasets, some of the predictive signal solely resides in text fields, while other predictive information is restricted to tabular feature values, or complex interactions between text and tabular values. To enjoy the benefits of end-to-end learning without sacrificing accuracy, we consider how to adapt a Transformer network to simultaneously operate on inputs from both modalities,

referring to the resulting network as *Multimodal-Net*. A natural approach in our setting is to enhance the Transformer such that its attention mechanism can contextualize representations of individual text tokens based not only on other parts of the text, but also on the values of relevant tabular features as well. Below we discuss three different options for implementing the *Multimodal-Net* that are depicted in Figure 1 (with details in Appendix A.3). These options differ in whether information is fused across text and tabular modalities: at the input layer (*All-Text*), in the earlier layers of the network near the input (*Fuse-Early*), or in the later layers of the network near the output (*Fuse-Late*).

**All-Text**   A simple (yet crude) option is to convert numeric and categorical values to strings and subsequently treat their columns also as text fields [52]. Through its byte-pair encoding, a pretrained Transformer can handle most categorical strings and may be able to crudely represent numeric values within a certain range (here we round all numbers to 3 significant digits in their string representation).

**Fuse-Early**   Rather than casting them as strings, we can allow our model to adaptively learn token representations for each numeric and categorical feature via backpropagation (see Figure 1b). We introduce an extra factorized embedding layer [26, 42] to map categorical values into the same $\mathbb{R}^d$ vector representation encoded by the pretrained Transformer backbone for text tokens (with different embedding layers used for different categorical columns in the table). All numeric features are encoded via a single-hidden-layer Multi-layer Perceptron (MLP) to obtain a unified $\mathbb{R}^d$ vector representation. The resulting $d$-dimensional vector representations from each modality are jointly fed into a 6-layer Transformer encoder whose self-attention operations can model interactions between the embeddings of text tokens, categorical values, and numeric values. We refer to this strategy as *Fuse-Early* because only a minimal (yet adaptive) input processing layer is added to convert the tabular features into a common vector form which can be jointly fed through many shared Transformer layers. Huang et al. [33] considered a similar strategy for applying Transformers to entirely numeric/categorical data, albeit without text components that are a major focus here.

**Fuse-Late**   Rather than aggregating information across modalities in early network layers, we can perform separate neural operations on each data type and only aggregate per-modality representations into a single representation near the output layer (see Figure 1c). This multi-branch design allows each branch to extract higher-level representations of the values from each modality, before the network needs to consider how modalities should be fused. Here we use a multi-tower architecture in which numeric and categorical features are fed into separate MLPs for each modality. The text features are fed into a (pretrained) Transformer network. The topmost vector representations of all three networks are pooled into a single vector (via either: mean/max pooling or concatenation) from which predictions are output via two dense layers.

## 5   Featurizing Text for Tabular Models

Despite their success for modeling text, the application of Transformer architectures to tabular data remains limited [17, 18, 33]. The use of tabular models together with Transformer-like text architectures has also received little attention [39, 63]. Note that 'tabular models' throughout are those trained on only numeric/categorical features, e.g. different types of decision tree ensembles.

In this paper, all tabular (numeric/categorical) modeling is simply done via AutoGluon-Tabular, an easy-to-use and highly accurate open-source tool for automated supervised learning on tabular data [4, 15, 18, 19, 70]. AutoGluon achieves strong performance by ensembling a diverse suite of high-quality models for tabular data, including: multiple variants of Gradient Boosted Decision Trees [9, 38, 49], Extremely Randomized Trees [24], and fully-connected Neural Networks (MLP) [15]. While neural networks are typically favored for unstructured data like text, decision tree ensembles have proven to be one of the most consistently performant models for tabular data [3, 18, 33]. While deploying home-grown ensembles can be tricky, AutoGluon automatically constructs and deploys its ensembles without any engineering overhead for the user. For real-time applications with latency constraints, AutoGluon provides many options to accelerate ensemble inference via pruning or distillation [18]. Since we have contributed the multimodal ensembling techniques of this paper into AutoGluon, our strategies can be utilized with all of the same benefits. Furthermore, AutoGluon optionally provides sophisticated hyperparameter-tuning [40] for all of its models, which can now be easily applied to our proposed text/tabular modeling pipeline as well.

To allow tabular models to access information in text fields, the text is typically first mapped to a continuous vector representation which replaces a text column in our data table with multiple numeric

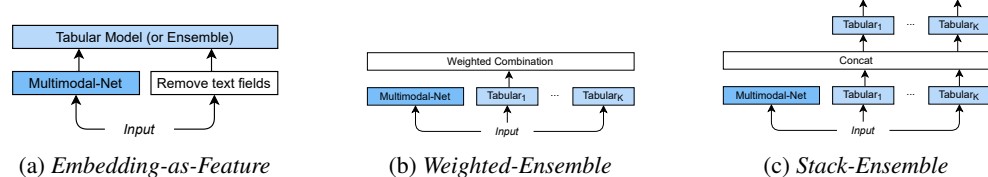

| (a) *Embedding-as-Feature* | (b) *Weighted-Ensemble* | (c) *Stack-Ensemble* |

Figure 2: Options for combining *Multimodal-Net* with classical tabular models. Five particular tabular models are used in this paper: extremely randomized trees, a simple MLP, and three different types of gradient boosted decision trees. Over our benchmark, option (c) performs the best and is chosen as the strategy for aggregating text and tabular models in our proposed AutoML solution.

columns (one for each vector dimension). One can treat each text column as a document, and each individual text field as a paragraph within the document, such that each text field can be featurized via NLP methods for computing text representations [14, 47, 54] before the tabular models are trained.

## 5.1 Neural Embedding of Text as Tabular Features

Rather than classical NLP methods like N-grams or word embeddings [14], a Transformer can instead be used to map the text fields into a vector representation via contextual embedding [5, 13]. Subsequently, the text fields are replaced in the data table by additional numeric columns corresponding to each dimension of the embedding vector (*Embedding-as-Feature* in Figure 2a). We consider three ways to featurize text using a Transformer.

**Pre-Embedding**  Most straightforward is to embed text via a pretrained Transformer (not fine-tuned on our labeled data), and subsequently train tabular models over the featurized data table [5].

**Text-Embedding**  The *Pre-Embedding* strategy is not informed about our particular prediction problem and the domain of the text data. In *Text-Embedding*, we further fine-tune the pretrained Transformer to predict our labels from only the text fields, and use the resulting Text-Net to embed the text. By adapting to the domain of the specific prediction task, *Text-Embedding* is able to extract more relevant textual features that can improve the performance of tabular models. This is particularly true in settings where the target only depends on one out of many text fields, since the fine-tuning process can produce representations that vary more based on the relevant field vs. irrelevant text.

**Multimodal-Embedding**  Text representations may improve when self-attention is informed by context regarding numeric/categorical features. Thus we also consider embedding text via our best multimodal network from Section 4.2 (depicted in Figure 1c). These models are again fine-tuned using the labeled data and now produce a single vector representation for *all* columns in the dataset, regardless of their type. Since Transformers are better suited for modeling text than tabular features, we only replace the text fields with the learned vector, all other non-text features are kept and used for subsequent tabular learning. Thus the sole difference between *Text-Embedding* and *Multimodal-Embedding* is that the embeddings used to replace text are additionally contextualized on numeric/categorical feature values in the latter method.

## 6 Aggregating Text & Tabular Models

Rather than merely leveraging the Transformers for their embedding vector representations as in Section 5.1, an alternative multimodal text/tabular modeling strategy is to instead consider their predictions and ensemble these with predictions from tabular models. Utilized by most AutoML frameworks [15, 21, 43], model ensembling is a straightforward technique to boost predictive accuracy. Ensembling is particularly suited for multimodal data, where different models may be trained with different modalities. However, the resulting ensemble may then be unable to exploit nonlinear predictive interactions between features from different modalities. To remedy this, we advocate for the use of our multimodal Transformers (from Section 4.2) that fuse information from text and tabular inputs. Furthermore, we propose stack ensembling with nonlinear aggregation of model predictions that can exploit inter-modality interactions between different base models' predictions, even when base models do not overlap in modality.

**Weighted-Ensemble**   We first consider straightforward aggregation via a weighted average of the predictions from our Transformer model and various tabular models (like those trained by AutoGluon-Tabular). Here, our Transformer and other models are independently trained using a common training/validation split. Subsequently, we apply *ensemble selection*, a forward-selection algorithm to fit aggregation weights over all models' predictions on the held-out validation data [8]. Unlike regression for fitting the aggregation weights [43, 60], ensemble selection is favored by many tabular AutoML tools like AutoGluon as it is more computationally efficient, less prone to overfitting, and naturally favors sparse weights [15, 22].

**Stack-Ensemble**   Rather than restricting the aggregation to a linear combination, we can use stacking [68]. This trains another ML model to learn the best aggregation strategy. The features upon which the 'stacker' model operates are the predictions output by all base models (including our Transformer), concatenated with the original tabular features in the data. Following Erickson et al. [15], we try each type of tabular model in AutoGluon-Tabular as a stacker model (see Appendix A.5). To output predictions, a weighted ensemble is constructed via ensemble selection applied to the tabular stacker models (Figure 2c). We do not consider our larger Transformer model as a stacker since lightweight aggregation models are preferred in practice. Overfitting is a key peril in stacking, and we ensure that stacker models are only trained over *held out* predictions produced from base models via 5-fold cross-validation (bagging) [15, 60].

# 7   Experiments

Here we empirically evaluate the many aforementioned multimodal AutoML strategies. To keep our study tractable, we adopt a sequential decision making process that decomposes the overall design into three stages: 1) determine the appropriate Transformer backbone and fine-tuning strategy for text data alone (Section 4.1), 2) determine the best way to extend this Transformer to text and tabular inputs (Section 4.2), and 3) choose the best method to combine text and tabular models (Sections 5 and 6). At each subsequent stage of the study, we explore modeling choices that are specific to that stage and simply use the best choice found in the empirical comparisons of the options available in previous stages. Myopic sequential design may fail to identify particularly synergistic choices across all stages of the AutoML pipeline as it favors choices which are independently performant at each stage and complement the choices made in previous stages. There is however no way to practically evaluate a larger combinatorial assortment of possible choices, and our sequential restriction may actually lead to a more robust/modular AutoML solution that better generalizes to new datasets with unique characteristics not found in our benchmark.

Each modeling strategy is run over our benchmark of 15 tabular datasets with text fields, detailed in Section 3. For straightforward comparison, we employ the most commonly used classification/regression evaluation metrics that are bounded in $[0, 1]$ with higher values indicating superior performance. We evaluate regression tasks via the coefficient of determination $R^2$, multiclass classification tasks via accuracy, and binary classification tasks via area under the ROC curve (AUC).

**Choice of Transformer Backbone**   Our first decision concerns the Transformer network itself, including what architecture and pretraining objective to employ. Existing results may not translate to our setting, since Transformers are typically applied to datasets with at most a couple text fields per training example [64, 65]. Here we choose between the (standard, already pretrained) base version of RoBERTa [45] or ELECTRA [10], two popular backbones used across modern NLP applications.

We first fine-tune the pretrained Transformer models as our sole predictors, using only the text features in each dataset. This helps identify which model is better at handling the types of text in our multimodal datasets. During fine-tuning of both of the RoBERTa or ELECTRA networks, we additionally consider two tricks to boost performance: 1) Exponentially decay the learning rate of the network parameters based on their depth [57]. We use a per-layer learning rate multiplier of $\tau^d$ in which $d$ is the layer depth and $\tau$ is the decay factor (set $= 0.8$ throughout). 2) Average the weights of the models loaded from the top-3 training checkpoints with the best validation scores [62].

The first section of Table 3 shows that ELECTRA performs better than RoBERTa across the text columns in our benchmark datasets. Our exponential decay and checkpoint-averaging tricks further boost performance, with the majority of additional gains produced by exponential decay. In subsequent experiments, we thus fix ELECTRA fine-tuned with both exponential decay and checkpoint-averaging as the model used to handle text features and call it *Text-Net*.

| Method | prod | qaq | qaa | cloth | airbnb | ae | mercari | jigsaw | imdb | fake | kick | jc | wine | pop | channel | avg ↑ | mrr ↑ |
|---|---|---|---|---|---|---|---|---|---|---|---|---|---|---|---|---|---|
| *Choosing Text-Net:* | | | | | NLP Backbones and Fine-tuning Tricks (Section 4.1) | | | | | | | | | | | | |
| RoBERTa | 0.588 | 0.412 | 0.268 | 0.700 | 0.344 | 0.953 | 0.561 | 0.960 | 0.731 | 0.929 | 0.751 | 0.615 | 0.811 | -0.000 | 0.301 | 0.595 | 0.07 |
| ELECTRA | 0.705 | 0.410 | 0.356 | 0.718 | 0.349 | 0.955 | 0.586 | 0.965 | 0.750 | 0.824 | 0.754 | 0.606 | 0.813 | 0.003 | 0.315 | 0.607 | 0.17 |
| + Exponential Decay $\tau = 0.8$ | 0.728 | 0.436 | 0.431 | 0.743 | 0.337 | 0.953 | 0.579 | 0.963 | 0.852 | 0.963 | 0.760 | 0.664 | 0.808 | 0.004 | 0.308 | 0.635 | 0.09 |
| + Average 3 ★ | 0.729 | 0.451 | 0.432 | 0.746 | 0.350 | 0.954 | 0.581 | 0.965 | 0.858 | 0.961 | 0.766 | 0.656 | 0.807 | 0.004 | 0.307 | 0.638 | 0.12 |
| *Choosing Multimodal-Net:* | | | | | Fusion Strategy (Section 4.2, Figure 1) | | | | | | | | | | | | |
| All-Text | 0.907 | 0.454 | 0.419 | 0.746 | 0.366 | 0.957 | 0.599 | **0.967** | 0.840 | 0.967 | **0.799** | 0.645 | 0.810 | 0.013 | 0.480 | 0.665 | 0.19 |
| Fuse-Early | **0.913** | 0.441 | 0.418 | 0.745 | 0.377 | 0.953 | 0.596 | **0.967** | 0.843 | 0.960 | 0.770 | 0.653 | 0.806 | 0.013 | 0.474 | 0.662 | 0.24 |
| Fuse-Late, Concat ★ | 0.907 | 0.449 | **0.445** | 0.747 | 0.395 | 0.958 | 0.603 | 0.966 | 0.857 | 0.961 | 0.773 | 0.639 | 0.812 | 0.015 | 0.481 | 0.667 | 0.17 |
| Fuse-Late, Mean | 0.912 | **0.458** | 0.431 | 0.748 | 0.399 | 0.955 | 0.602 | **0.967** | 0.869 | 0.963 | 0.773 | 0.625 | 0.807 | 0.015 | 0.478 | 0.667 | 0.09 |
| Fuse-Late, Max | 0.910 | 0.452 | 0.429 | 0.747 | 0.401 | 0.956 | 0.599 | 0.966 | 0.863 | 0.957 | 0.761 | 0.634 | 0.808 | 0.015 | 0.484 | 0.665 | 0.12 |
| *Choosing Aggregation:* | | | | | Multimodal Model Aggregation (Sections 5.1 and 6, Figure 2) | | | | | | | | | | | | |
| Pre-Embedding | 0.895 | 0.216 | 0.247 | 0.642 | 0.449 | 0.972 | 0.433 | 0.586 | 0.871 | 0.926 | 0.743 | 0.491 | 0.680 | 0.012 | 0.526 | 0.579 | 0.13 |
| Text-Embedding | 0.867 | 0.446 | 0.432 | 0.748 | 0.430 | 0.972 | 0.434 | 0.587 | 0.855 | 0.962 | 0.790 | 0.658 | 0.830 | 0.008 | 0.502 | 0.635 | 0.20 |
| Multimodal-Embedding | 0.907 | 0.439 | 0.437 | 0.749 | 0.438 | 0.974 | 0.432 | 0.587 | 0.847 | 0.967 | 0.794 | **0.683** | 0.829 | 0.007 | 0.517 | 0.640 | 0.18 |
| Weighted-Ensemble | 0.907 | 0.439 | 0.429 | 0.744 | 0.453 | 0.976 | 0.597 | 0.957 | 0.876 | 0.923 | 0.787 | 0.641 | 0.814 | 0.018 | 0.554 | 0.674 | 0.39 |
| Stack-Ensemble ★ | 0.909 | 0.456 | 0.438 | 0.751 | 0.459 | 0.977 | **0.605** | **0.967** | **0.878** | 0.964 | 0.797 | 0.624 | 0.836 | **0.020** | **0.556** | **0.683** | **0.59** |
| | | | | | Tabular AutoML + Feature Engineering Baselines (Section 5) | | | | | | | | | | | | |
| AG-Weighted | 0.891 | 0.046 | 0.076 | -0.002 | 0.426 | 0.841 | 0.098 | 0.587 | 0.845 | | 0.668 | 0.004 | 0.173 | 0.016 | 0.549 | 0.394 | 0.11 |
| AG-Stack | 0.891 | 0.046 | 0.077 | 0.001 | 0.435 | 0.841 | 0.098 | 0.587 | 0.844 | 0.697 | 0.670 | 0.003 | 0.175 | 0.017 | 0.550 | 0.395 | 0.10 |
| AG-Weighted+ N-Gram | 0.892 | 0.426 | 0.382 | 0.610 | 0.450 | 0.978 | 0.526 | 0.909 | 0.842 | 0.966 | 0.772 | 0.357 | 0.829 | 0.019 | 0.546 | 0.633 | 0.11 |
| AG-Stack+ N-Gram | 0.895 | 0.414 | 0.383 | 0.654 | **0.466** | **0.979** | 0.569 | 0.915 | 0.850 | **0.968** | 0.775 | 0.612 | **0.842** | **0.020** | 0.548 | 0.659 | 0.19 |
| H2O AutoML | 0.869 | 0.247 | 0.159 | 0.163 | 0.329 | 0.976 | 0.430 | 0.531 | 0.813 | 0.756 | 0.669 | 0.411 | 0.478 | 0.014 | 0.530 | 0.492 | 0.11 |
| H2O AutoML + Word2Vec | 0.859 | 0.244 | 0.285 | 0.624 | 0.347 | 0.973 | 0.534 | 0.847 | 0.827 | 0.943 | 0.755 | 0.443 | 0.778 | 0.013 | 0.524 | 0.600 | 0.16 |
| H2O AutoML + Pre-Embedding | 0.846 | 0.227 | 0.312 | 0.644 | 0.367 | 0.969 | 0.282 | 0.572 | 0.874 | 0.893 | 0.738 | 0.549 | 0.571 | 0.007 | 0.501 | 0.557 | 0.12 |

Table 3: Accuracy (and $R^2$, AUC) of AutoML strategies over our multimodal benchmark. Column **avg** lists each method's average score across datasets (i.e. how *much* methods differ in overall performance) and **mrr** its mean reciprocal rank among all evaluated methods (i.e. how *often* a method outperforms others). Each subsection encapsulates a design stage (★ marks variant with best avg).

**Best Multimodal Network**   Next, we explore the best way to extend the *Text-Net* model to operate across numeric/categorical inputs in addition to text fields. Three multimodal network variants are considered here: *All-Text*, *Fuse-Early*, *Fuse-Late* (see Figure 1). Across our datasets, Table 3 shows that the *Fuse-Late* strategy outperforms the other options for producing predictions from multimodal inputs using a single neural network (including *Text-Net*). We thus fix this model as our *Multimodal-Net* used in subsequent experiments.

**Aggregating Transformers and Tabular Models**   Having identified a good neural network architecture for multimodal text/tabular inputs, we now study combinations of such models with classical learning algorithms for tabular data. Where not specified, the tabular models are those trained by AutoGluon-Tabular (see Appendix A.5). Here we considered the following aggregation strategies: *Pre-Embedding*, *Text-Embedding*, *Multimodal-Embedding*, *Weighted-Ensemble*, *Stack-Ensemble*.

The third section of Table 3 illustrates that *Stack-Ensemble* is overall the best aggregation strategy. As expected, *Text-Embedding* and *Multimodal-Embedding* outperform *Pre-Embedding*, demonstrating how domain-specific fine-tuning improves the quality of learned embeddings. *Multimodal-Embedding* performs better than *Text-Embedding* on some datasets and similarly across the rest, showing it can be beneficial to use text representations contextualized on numeric/categorical information.

**AutoGluon Baselines**   As most of our results are based around the tabular models in AutoGluon [15], we also compare different variants of AutoGluon (without our *Multimodal-Net*) as baselines:

*AG-Weighted / AG-Stack*: We train AutoGluon with weighted / stack ensembling of its tabular models, here ignoring all text columns.

*AG-Weighted + N-Gram / AG-Stack + N-Gram*: Similar to *AG-Weighted / AG-Stack*, except we first use AutoGluon's N-Gram featurization [14] to encode all text in tabular form.

The performance gap between AutoGluon-Tabular with and without N-Grams can reveal (an approximate lower bound for) how much extra predictive value is provided by the text features in each dataset. Inspecting these gaps, we find that, compared to the tabular features, text features contain most of the predictive signal in some datasets (qaq, qaa, cloth, mercari, jc), and far less signal in other datasets (prod, imdb, channel). Note that our proposed *Stack-Ensemble* performs relatively well across all types of datasets, regardless how the predictive signal is allocated between text and tabular features.

**H2O Baselines**   In addition to AutoGluon, we also run another popular open-source AutoML tool offered by H2O. Since H2O AutoML is not designed for the text in our multimodal data tables, we try combining H2O's NLP tool [29] and tabular AutoML tool [43].

*H2O AutoML*: We run H2O AutoML directly on the original data of our benchmark. It is assumed that H2O AutoML ignores all text features (as a tabular AutoML framework), but H2O categorizes

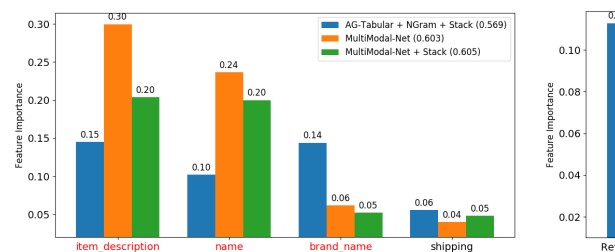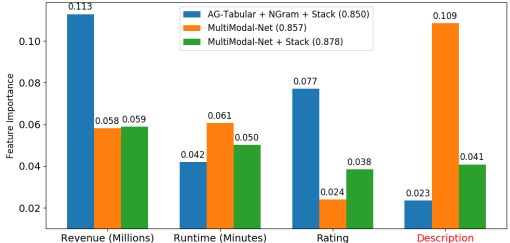



(a) Permutation importance in "mercari".      (b) Permutation importance in "imdb".

Figure 3: Importance of text vs. tabular features for three models in two datasets (text features in red).



text vs. other feature types slightly differently than us.

*H2O AutoML + Word2Vec*: We run H2O's word2vec algorithm to featurize text fields and then H2O AutoML on the featurized data, following their recommended procedure [29].

*H2O AutoML + Pre-Embedding*: We featurize each text field using embeddings from a pretrained ELECTRA Transformer, as in *Pre-Embedding*, followed by H2O AutoML on the featurized data table.

The last section of Table 3 shows that while these powerful AutoML ensemble predictors can outperform our individual neural network models (particularly for datasets with more tabular-signal), our proposed *Stack-Ensemble* and *Weighted-Ensemble* are superior overall. Given the success of pretrained Transformers across NLP, we are surprised to find both N-Grams and word2vec here provide superior text featurization than *Pre-Embedding*.

**Performance in Real-world ML Competitions**   Some datasets in our multimodal benchmark originally stem from ML competitions. For these (and other recent competitions with text/tabular data), we fit our automated solution using the official competition dataset, without manual adjustment or data preprocessing. We then submit its resulting predictions on the competition test data to be scored, which enables us to see how they fare against the manual efforts of human data science teams.

Our *Stack-Ensemble* model achieves 1st place historical leaderboard rank in two MachineHack prediction competitions: *Product Sentiment Classification*[3] and *Predict the Data Scientists Salary in India*[4], and this model achieves 2nd place in another: *Predict the Price of Books*[5], as well as a Kaggle competition: *California House Prices*[6]. Simply training only our *Multimodal-Net* suffices to achieve 2nd place in a very popular Kaggle competition in which 2380 teams participated: *Mercari Price Suggestion Challenge*[7] (which offered a $100,000 prize). These results demonstrate that, without any manual adjustment, the AutoML strategy identified from our benchmark is competitive with human data scientists on real-world text/tabular datasets that possess great commercial value.

**Feature Importance Analysis**   Feature importance helps us understand what drives a ML system's accuracy and whether text fields in a dataset are worth their overhead. We compute permutation feature importance [6] for our models, which is defined as the drop in prediction accuracy after values of only this feature (which are entire text fields for a text column) are shuffled in the test data (across rows). We only shuffle original column values so our importance scores are not biased by preprocessing/featurization decisions (except in how these directly affect model accuracy). Figure 3 shows that both our *Multimodal-Net* and *Stack-Ensemble* containing this network rely more heavily on text features than the *AG-Stack+N-Gram* baseline. With more powerful modeling of text fields, models often begin to rely more heavily on the text fields. An exception here is the *brand_name* feature in mercari, but this feature usually contains just a single word in its fields.

---

[3] https://www.machinehack.com/hackathons/product_sentiment_classification_weekend_hackathon_19/overview ("Anonymous Submission ID 1556" entry)

[4] https://machinehack.com/hackathons/predict_the_data_scientists_salary_in_india_hackathon/overview ("Xingjian Shi" entry)

[5] https://machinehack.com/hackathons/predict_the_price_of_books/overview

[6] https://www.kaggle.com/c/california-house-prices ("sxjscience" entry)

[7] *Multimodal-Net* achieved a score of 0.38685 on the private leaderboard: https://github.com/submission001/anonymoussubmission_automl/blob/master/competition_submissions/mercari_submission_screenshot.png.

# 8 Discussion

Lacking public benchmarks, academic research on ML for multimodal text/tabular data has not matched industry demand to derive practical value from such data. This paper provides evidence that generic best practices for such data remain unclear today: we simply evaluated a few basic strategies on our benchmark and found a single automated strategy that turns out to outperform top human data scientists in numerous historical prediction competitions involving diverse text/tabular data. This strategy uses a stack ensemble (Section 6) of tabular models trained on top of predictions from other tabular models and a *Multimodal-Net* (depicted in Figure 2c). The latter network is based on a *Fuse-Late* architecture (depicted in Figure 1c) with concatenation of text, numeric, and categorical representations (where text representations are produced via the ELECTRA Transformer backbone) and is trained via fine-tuning with exponential learning rate decay and checkpoint averaging. The competitive performance of this empirically-identified strategy supports the premise that our benchmark is sufficiently diverse and representative of real-world text/tabular prediction tasks. Our rigorous benchmark challenges conventional beliefs:

- Neural embedding of text followed by tabular modeling (*Pre/Text-Embedding*) [5, 29] is often outperformed by N-gram featurization (*AG-Stack + N-Gram*) or leveraging predictions from text neural networks (*Stack-Ensemble*) rather than their representations (embeddings).

- In the architecture of multimodal networks for classification/regression, newer ideas to fuse modalities in early layers (i.e. *Fuse-Early/All-Text* Transformers with cross-modality attention [32, 52, 55]) are not necessarily superior to older multi-tower *Fuse-Late* architectures that fuse representations in higher layers closer to the output [2, 36, 53].

- An end-to-end multimodal neural network is surpassed by stack ensembling this *Multimodal-Net* with tabular models trained in separate stages rather than end-to-end (*Stack-Ensemble*).

Previously anticipated conclusions that are empirically validated by our benchmark include:

- Text featurization is better via fine-tuned networks (*Text-Embedding*) than pretrained ones (*Pre-Embedding*), and slightly better via a fine-tuned multimodal network (*Multimodal-Embedding*), whose text embeddings benefit from contextualization on the tabular features.

- Naively casting numeric/categorical features as strings (*All-Text*) is simple yet effective [52].

- Able to exploit predictive interactions between different modalities, stack ensembling outperforms simple weighted ensembling, yet it still facilitates modular system design.

Further analysis of our benchmark can reveal many more practical ML insights. Important questions not considered here include *how to best*: Handle many long text fields? Perform multimodal feature selection? Apply feature engineering that combines synergistically with learned neural network representations? Allocate limited training/HPO time between cheaper tabular models and more expensive text neural networks? We consider the study presented in this paper as a starting point for multimodal AutoML with text/tabular data. welcome contributions to improve them further. Our public benchmark and open-source methods will hopefully stimulate the AutoML community to broaden the applicability of their methods to more heterogeneous data types, especially those modalities that commonly co-occur in real-world ML applications.

We caution our benchmark only contains text in the English language and primarily from commercial domains. Thus its conclusions will only hold for particular types of applications. To ensure similar advancements for text/tabular data with low-resource languages [31, 37, 41], we encourage the development of a similar benchmark with non-English text. We also caution that analysis of text fields may raise privacy concerns as such fields may expose arbitrary personal information [7, 20]. Since text fields may contain arbitrary information, they are also prone to introducing spurious correlations in training data that may harm accuracy during deployment [59] and may be undesirably coupled to protected attributes such as race, gender, or socioeconomic status [67]. Basing automated business decisions on customer-generated text could also be more susceptible to adversarial manipulation [44] than tabular features that customers cannot as easily control.

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
