# OpenReview forum: "Multimodal AutoML on Tables with Text Fields"
_NeurIPS.cc/2021/Track/Datasets_and_Benchmarks/Round1 — Submitted to NeurIPS 2021 Datasets and Benchmarks Track (Round 1)_

### Official Review · Reviewer_4Cq5 · 2021-06-29
**This paper introduces a benchmark assembled from the public datasets for evaluating multimodal (tabular/text) models.**

**Rating:** 6
**Confidence:** 3
**Correctness:** Yes, the benchmark, datasets and desc…
**Clarity:** This paper is clearly written

**Strengths:**

1. This multimodal text/tabular benchmark is relevant for the research community and industry.
2. Benchmark methodology is clearly described including dataset creation, evaluated models, hyperparameters, tradeoffs etc.
3. Results regarding Transformer networks can be useful for the community.
4. Two AutoML solutions are presented with relevant ablation study.

**Weaknesses:**

1. There is a certain confusion about positioning of this paper: as a novel benchmark or a new multimodal AutoML method. For example, the AutoML title and some parts of the paper may confuse readers regarding the paper scope. I'd suggest authors to revise parts of the paper in order to clearly position it as a multimodal benchmark to be aligned with the goals of this NeurIPS track. It will also bring more attention to this benchmark from the research community.

2. I couldn't verify Mercari Price Suggestion Challenge leaderboard results on Kaggle.

**Additional Feedback:**

This paper, if clearly positioned as a new multimodal tabular/text benchmark rather than AutoML method, can be very relevant to the broader research community. I would further increase my score if weakness # 1 is addressed.

Upd. 07/15: I appreciate author's feedback. However, I intend not to increase my score, because the current revision is still written using a half-benchmark and half-analysis style. Then, the analysis part is biased towards an AutoML framework being associated with author affiliation, which can limit research community interest. While benchmark is a compilation of the existing datasets,  I believe, it is very relevant to this NeurIPS track, if authors organize their paper solely around the benchmark itself rather than particular AutoML framework.

**Documentation:**

Yes, code and documentation are provided

**Relation To Prior Work:**

Yes, prior works on multimodal benchmarks are clearly described

**Summary And Contributions:**

In this paper, authors introduce a new benchmark that has been assembled from 15 existing public datasets. This benchmark goal is to evaluate models in multimodal (tabular/text data) setting using real-world data. In addition, authors analyze NLP embedding extractors, multimodal architectures and AutoML solutions to improve results on their benchmark. Authors clearly describe their methodology for constructing datasets, benchmarking and model comparisons. The code is available and has documentation how to reproduce all experiments.

---

> ### Author Response · Authors · 2021-07-15
> **Response to Reviewer 4Cq5 - Part 1**
>
> Thank you for the feedback and the favorable review. We are glad you thought: the benchmark is relevant for the research community and industry, and our benchmark methodology is clearly described including dataset creation, evaluated models, hyperparameters, tradeoffs etc. We address your specific questions/concerns below.
>
>
> > There is a certain confusion about positioning of this paper: as a novel benchmark or a new multimodal AutoML method. For example, the AutoML title and some parts of the paper may confuse readers regarding the paper scope. I'd suggest authors to revise parts of the paper in order to clearly position it as a multimodal benchmark to be aligned with the goals of this NeurIPS track. It will also bring more attention to this benchmark from the research community. This paper, if clearly positioned as a new multimodal tabular/text benchmark rather than AutoML method, can be very relevant to the broader research community. I would further increase my score if weakness # 1 is addressed.
>
> We note the CFP for this Track states: "systematic analyses of existing systems on novel datasets that yield important new insight are also in scope". Our paper presents both a new benchmark (set of tabular+text  datasets) as well as a systematic analysis of various strategies for handling tabular+text data. We now clarify in the Introduction section:
>
> "Note that we write AutoML to describe any modeling strategy that is robustly performant across a diverse set of datasets without manual adjustments. The AutoML method promoted in this paper is simply the strategy that happened to perform best in our systematic analysis of various straightforward strategies over the proposed benchmark.  It is the development of our benchmark and our systematic analysis that directly led to this AutoML method, whose efficacy was subsequently verified in historical prediction competitions (demonstrating that our benchmark and analysis have led to important new insights)."
>
> Furthermore, we have rewritten parts of the paper to emphasize the benchmark far more, as well as the fact that the methodological section is really describing a systematic analysis of popular strategies rather than an entirely novel AutoML methodology. Appendix B now contains much more material regarding dataset/benchmark descriptions. The "AutoGluon Baselines" subsection now contains  additional benchmark insights: "The  performance  gap  between  AutoGluon-Tabular  with  and  without  N-Grams  can  reveal  (an approximate lower bound for) how much extra predictive value is provided by the text features in each dataset. Inspecting these gaps, we find that, compared to the tabular features, text features contain most of the predictive signal in some datasets (qaq, qaa, cloth, mercari, jc), and far less signal in other datasets (prod, imdb, channel). Note that our  proposed Stack-Ensemble performs relatively well across all types of datasets, regardless how the predictive signal is allocated between text and tabular features."
>
> In "Benchmarking Multimodal Text/Tabular AutoML" section, we have also added an expanded  paragraph to emphasize the benchmark further:
> "Our benchmark strives to represent the types of ML tasks that commonly arise in industry today. Appendix B provides detailed descriptions of each dataset. In creating the benchmark, we aimed to include a mix of classification vs. regression tasks and datasets from real applications (as opposed to toy academic settings) that contain a rich mix of text, numeric, and categorical columns. Table 2 shows it is comprised of datasets that are quite diverse in terms of: sample-size, problem types, number of features, and type of features. 11 of the datasets contain more than one text field (with 28 text fields in the airbnb dataset). These text fields greatly vary in the amount of text they contain (e.g. short product names vs. lengthy product descriptions/reviews). The data (and text vocabulary) stem from a mix of of real-world domains spanning:  e-commerce, news, social media, question-answering, and product listings (jobs, projects, films, Airbnb).  Subsequent accuracy results from Table 3 indicate the 15 underlying prediction problems also vary greatly in terms of both difficulty and how the predictive signal is divided between text/tabular modalities. To reflect real-world ML issues, we processed the data minimally (beyond ensuring the features/labels correspond to meaningful prediction tasks without duplicate examples) and thus there are arbitrarily-formatted strings and missing values all throughout. Systems that can perform well across the diverse set of 15 benchmark datasets are thus likely to provide real-world value for an important class of applications."
>
> We hope our significant revisions have strengthened the paper in the reviewer's eyes and they will increase their score as initially mentioned.

---

> > ### Author Response · Authors · 2021-07-15
> > **Response to Reviewer 4Cq5 - Part 2**
> >
> > > I couldn't verify Mercari Price Suggestion Challenge leaderboard results on Kaggle.
> >
> > We have updated the code repository to include the instructions for reproducing our submissions to "Kaggle: Mercari Price Suggestion Challenge"  (https://github.com/submission001/anonymoussubmission_automl/tree/master/competition_submissions#mercari-price-suggestion-challenge) and "MachineHack: Product Sentiment Analysis" (https://github.com/submission001/anonymoussubmission_automl/tree/master/competition_submissions#product-sentiment-analysis). In addition, we have attached the screenshot that shows our submission achieved a score of 0.38685 on the private leaderboard  (https://github.com/submission001/anonymoussubmission_automl/blob/master/competition_submissions/mercari_submission_screenshot.png).

---

### Official Review · Reviewer_eEQg · 2021-07-06
**Interesting paper that builds on a set of datasets for free text + tabular data but could describe the datasets in more detail**

**Rating:** 7
**Confidence:** 3
**Correctness:** The claims look correct to me.
**Clarity:** Could improve clarity as discussed ab…

**Strengths:**

The authors do a good job of trying out many combinations of methods and also show experiments from another AutoML framework (H2O) apart from their own. The ideas and motivations behind the methods seem good to me. They also divide the design stages into 3 logical stages and perform experiments and give interesting insights in each stage.


**Weaknesses:**

The main weakness of this paper to me seems that it describes the methods in much more detail than the the benchmark itself. It could describe and provide examples of some text fields in the benchmark that it feels would be problematic for one method and not for the other. It could also describe some of the winning solutions on datasets from competitions and why that particular dataset is useful for this benchmark, e.g., along some dimensions of difficulty. For example, it says
"However existing methods to input numeric/categorical features into Transformer models remain quite rudimentary [51] and fail to outperform the best tree algorithms for tabular prediction" and "In contrast, multiple prediction competitions each involving a single real-world text/tabular dataset have been held, but winning solutions have heavily relied on dataset/domain-specific tricks [45]" but doesn't provide too many details.

Another weakness is the organisation and explanation of some of the approaches. Section 4 and 5 I feel are not optimally organised. I think it could be better organised into 3 sections following the 3 design stages as defined in L231. When initially reading from L231 onwards, I felt the 2nd design stage there might have been pertaining to section 5.1 but I only later understood that it was referring to Section 4 and Figure 1.

Further in section 4, is Figure 1 only for the text under "Neural Architectures for Multimodal Data"?
Does the text under "Neural Architectures for Multimodal Data" combine the previous parts, i.e., "Featurizing Text for Tabular Models" and "Transformer Models for Text"?

I also feel "multimodal data" was vaguely defined and could be defined more precisely earlier on in the text.

I also didn't feel that the datasets were very large with the largest datasets being about 125,000 samples. I had expected a million or more for some of the datasets.

Some more weaknesses are covered later.

**Additional Feedback:**

L66: Could you please mention sizes of the cited "small academic datasets" for comparison? Because I felt that the largest of the datasets in the paper being 125,000 at most wasn't very large either.


L124: "One can treat each text column as a document, and each individual text field as a paragraph within the document, such that each text field can be featurized via NLP methods for computing text representations"

I'm not sure I understood the difference between text field and column.


L161: It was not clear to me what pretrained NLP model is meant here.

L200: I'm not sure which "previous multimodal networks" are referred to here.

L237: Did you mean Table 3?

L238: How were the evaluation metrics chosen?

L247: Were the fine-tuning tricks applied to Roberta or Electra?

How were mercari and IMDB chosen for figure 3? Is the figure representative of importances on all datasets? I think a scalar metric that summarises something like mean increase in importance for the text features across the datasets would be interesting.

L333: In the bulleted list beginning here, it took me quite some time to parse which exact results were being referred to in Table 3. Could you please also mention the method name from Table 3 in these bulleted lists?
For example, "Neural embedding of text followed by tabular modeling".
Did you mean to refer to the last line of Table 3? I think I can see other lines in the table that might have been meant as well.

L656: "we found performance did not qualitatively differ with other reasonable hyperparameter settings."

What do the authors mean by other reasonable hyperparameter settings?

L572: "The data were all already publicly available in different forms, and are not offensive."

To me it seems, the jigsaw dataset may contain offensive material.

In the checklist, mentioning exact Appendix sections would be better than just saying "See Appendix".


**Documentation:**

The documentation generally looks good to me.

**Ethics:**

L364: "We also caution that analysis of text fields raises particular privacy concerns as such fields tend to be unstructured and may expose arbitrary personal information."

This is the only line regarding ethical concerns. I'm sure some more ethical and fairness concerns could have been raised had the datasets been described in more detail.


**Relation To Prior Work:**

The authors discuss several previous related works.

**Summary And Contributions:**

(Note: This was an emergency review)
The authors collect together datasets containing different modalities of data (text and tabular) from various competitions and industry sources and, I believe, process them to be useable from a common API. They claim there is a lack of a common benchmark for this setting even though it's a very important industry use-case. They further design and evaluate new algorithms that test many design choices. Since different kinds of existing algorithms perform well on text vs tabular data (Transformers on text vs various algorithms in, e.g., AutoGluon, on tabular data), the authors' approaches seek to build baselines and algorithms that combine the strengths of different existing algorithms and evaluate design choices involved here. They hope to kickstart more ML research in the multimodal area.

---

> ### Author Response · Authors · 2021-07-15
> **Response to Reviewer eEQg - Part 1**
>
> Thank you for the feedback and the favorable review. We are glad you thought: the paper does a good job of trying out many combinations of methods, the ideas and motivations behind the methods seem good, and our experiments give interesting insights. We address your specific questions/concerns below.
>
>
> > The main weakness of this paper to me seems that it describes the methods in much more detail than the the benchmark itself. It could describe and provide examples of some text fields in the benchmark that it feels would be problematic for one method and not for the other. It could also describe some of the winning solutions on datasets from competitions and why that particular dataset is useful for this benchmark, e.g., along some dimensions of difficulty. For example, it says "However existing methods to input numeric/categorical features into Transformer models remain quite rudimentary [51] and fail to outperform the best tree algorithms for tabular prediction" and "In contrast, multiple prediction competitions each involving a single real-world text/tabular dataset have been held, but winning solutions have heavily relied on dataset/domain-specific tricks [45]" but doesn't provide too many details.
>
> We note the CFP for this Track states: "systematic analyses of existing systems on novel datasets that yield important new insight are also in scope". Our paper presents both a new benchmark (set of tabular+text  datasets) as well as a systematic analysis of various strategies for handling tabular+text data. We now clarify in the Introduction section:
>
> "Note that we write AutoML to describe any modeling strategy that is robustly performant across a diverse set of datasets without manual adjustments. The AutoML method promoted in this paper is simply the strategy that happened to perform best in our systematic analysis of various straightforward strategies over the proposed benchmark.  It is the development of our benchmark and our systematic analysis that directly led to this AutoML method, whose efficacy was subsequently verified in historical prediction competitions (demonstrating that our benchmark and analysis have led to important new insights)."
>
> Furthermore, we have rewritten parts of the paper to emphasize the benchmark far more, as well as the fact that the methodological section is really describing a systematic analysis of popular strategies rather than an entirely novel AutoML methodology. Appendix B now contains much more material regarding dataset/benchmark descriptions. The "AutoGluon Baselines" subsection now contains  additional benchmark insights: "The  performance  gap  between  AutoGluon-Tabular  with  and  without  N-Grams  can  reveal  (an approximate lower bound for) how much extra predictive value is provided by the text features in each dataset. Inspecting these gaps, we find that, compared to the tabular features, text features contain most of the predictive signal in some datasets (qaq, qaa, cloth, mercari, jc), and far less signal in other datasets (prod, imdb, channel). Note that our  proposed Stack-Ensemble performs relatively well across all types of datasets, regardless how the predictive signal is allocated between text and tabular features."
>
> In "Benchmarking Multimodal Text/Tabular AutoML" section, we have also added an expanded  paragraph to emphasize the benchmark further:
> "Our benchmark strives to represent the types of ML tasks that commonly arise in industry today. Appendix B provides detailed descriptions of each dataset. In creating the benchmark, we aimed to include a mix of classification vs. regression tasks and datasets from real applications (as opposed to toy academic settings) that contain a rich mix of text, numeric, and categorical columns. Table 2 shows it is comprised of datasets that are quite diverse in terms of: sample-size, problem types, number of features, and type of features. 11 of the datasets contain more than one text field (with 28 text fields in the airbnb dataset). These text fields greatly vary in the amount of text they contain (e.g. short product names vs. lengthy product descriptions/reviews). The data (and text vocabulary) stem from a mix of of real-world domains spanning:  e-commerce, news, social media, question-answering, and product listings (jobs, projects, films, Airbnb).  Subsequent accuracy results from Table 3 indicate the 15 underlying prediction problems also vary greatly in terms of both difficulty and how the predictive signal is divided between text/tabular modalities. To reflect real-world ML issues, we processed the data minimally (beyond ensuring the features/labels correspond to meaningful prediction tasks without duplicate examples) and thus there are arbitrarily-formatted strings and missing values all throughout. Systems that can perform well across the diverse set of 15 benchmark datasets are thus likely to provide real-world value for an important class of applications."

---

> > ### Author Response · Authors · 2021-07-15
> > **Response to Reviewer eEQg - Part 1 continued**
> >
> > Regarding dataset/domain-specific tricks: we have added some examples of these in Appendix B dataset descriptions. For mercari:
> > "This data originates from a Kaggle competition, in which the 1st place (https://www.kaggle.com/c/mercari-price-suggestion-challenge/discussion/50256) and 3rd place (https://www.kaggle.com/c/mercari-price-suggestion-challenge/discussion/50272) teams engineered dataset-specific text features such as customized bag-of-words (https://www.kaggle.com/whitebird/mercari-price-3rd-0-3905-cv-at-pb-in-3300-s\#L362-L365) and character N-grams from names, carefully-tuned learning-rate schedules (the 1st place solution doubles the batch-size for each epoch), and model ensembles that appear specifically crafted for just this dataset. Instead, the automated solution described in our paper requires very little human-engineering and generalizes to different types of multimodal text+tabular datasets."
> >
> > For jigsaw:
> > "This data originates from a Kaggle competition in which the 1st place solution (https://www.kaggle.com/c/jigsaw-unintended-bias-in-toxicity-classification/discussion/103280)  utilized dataset-specific tricks such as a Bucket Sequencing Collator, auxiliary domain-specific prediction tasks for models, and a custom mimic loss function for training. The 2nd place solution\footnote (https://www.kaggle.com/c/jigsaw-unintended-bias-in-toxicity-classification/discussion/100661) also used a custom loss function, and the 3rd place solution (https://www.kaggle.com/c/jigsaw-unintended-bias-in-toxicity-classification/discussion/97471) used custom target weighting, unlike our proposed AutoML solution which does utilize dataset-specific tricks."

---

> > > ### Author Response · Authors · 2021-07-15
> > > **Response to Reviewer eEQg - Part 2**
> > >
> > > > Another weakness is the organisation and explanation of some of the approaches. Section 4 and 5 I feel are not optimally organised. I think it could be better organised into 3 sections following the 3 design stages as defined in L231. When initially reading from L231 onwards, I felt the 2nd design stage there might have been pertaining to section 5.1 but I only later understood that it was referring to Section 4 and Figure 1. Further in section 4, is Figure 1 only for the text under "Neural Architectures for Multimodal Data"? Does the text under "Neural Architectures for Multimodal Data" combine the previous parts, i.e., "Featurizing Text for Tabular Models" and "Transformer Models for Text"?
> > >
> > > We have greatly improved the section order, connections between sections, and clarity regarding the proposed method concluded to be superior. In particular, the order in which the methodology is described in the newly rewritten Sections 4-6 now aligns with the 3 sequential design stages considered in our experiments (as nicely suggested by the reviewers). Here we have moved the "Featurizing Text for Tabular Models" to be its own section together with the "Neural Embedding of Text as Tabular Features" for a more unified presentation of ideas throughout the new Sections 4-6. Furthermore, we have added clearly motivated connection sentences to transition between each section and motivate the next section. In the Experiments section and Table 3, we now explicitly list the relevant method sections/figures when discussing each design stage to make it clear what is under consideration in each empirical comparison.
> > >
> > > To clear up any confusion regarding the method we regard to be superior, the caption of each figure now explicitly states what choice was the official one we used in our proposed AutoML solution.
> > > We also stated this choice explicitly at the end of the Introduction section:
> > > "The AutoML method promoted in this paper (stack ensembling of tabular models with a multimodal Transformer network) ...".
> > > And we restated this choice again in full detail in the Discussion section:
> > > "we simply evaluated a few basic strategies on our benchmark and found a single automated strategy that turns out to outperform  top human data scientists in numerous historical prediction competitions involving diverse text/tabular data. This strategy uses a stack ensemble (Section 6) of tabular models trained on top of predictions from other tabular models and a Multimodal-Net (depicted in Figure 2c). The latter network is based on a Fuse-Late architecture (depicted in Figure 1c) with concatenation of text, numeric, and categorical representations (where text representations are produced via the ELECTRA Transformer backbone) and is trained via fine-tuning with exponential learning rate decay and checkpoint averaging."
> > >
> > > In "Extending Transformer Architectures to Multimodal Inputs" section, we have defined Multimodal-Net early on to clarify that Figure 1 is specific to this section and have also added clarification:  "Below we discuss three different options for implementing the Multimodal-Net that are depicted in Figure 1 (with details in Appendix A.3). These options differ in whether information is fused across text and tabular modalities: at the input layer (All-Text), in the earlier layers of the network near the input (Fuse-Early), or in the later layers of the network near the output (Fuse-Late)."
> > >
> > > Finally, note that when we consider featurizing text fields via our network, the embeddings may be extracted from either the text-only network ("Text-Embedding") or from the multimodal network ("Multimodal-Embedding"). We have added clarification: "the sole difference between Text-Embedding and Multimodal-Embedding is that the embeddings used to replace text are additionally contextualized on numeric/categorical feature values in the latter method."
> > >
> > >
> > > > L333: In the bulleted list beginning here, it took me quite some time to parse which exact results were being referred to in Table 3. Could you please also mention the method name from Table 3 in these bulleted lists? For example, "Neural embedding of text followed by tabular modeling". Did you mean to refer to the last line of Table 3? I think I can see other lines in the table that might have been meant as well.
> > >
> > > In the bulleted list in Discussion section, we have now listed Table 3 method names in the discussion to eliminate the ambiguity.
> > >
> > >
> > > > I also feel "multimodal data" was vaguely defined and could be defined more precisely earlier on in the text.
> > >
> > > We have clarified in Introduction (2nd paragraph) that we use "multimodal" or "text/tabular" to refer to data tables that jointly contain numeric/categorical and text features.

---

> > > > ### Author Response · Authors · 2021-07-15
> > > > **Response to Reviewer eEQg - Part 3**
> > > >
> > > > > I'm not sure I understood the difference between text field and column.
> > > >
> > > > In Introduction section, we have clarified the setup/terminology more clearly:
> > > > "This paper considers tables of this form where rows contain IID training examples (each with a single numeric/categorical value to predict, i.e. regression/classification) and the columns used as predictive features can contain text, numeric, or categorical values. We refer to the value in a particular row and column as a field, where a single text field may actually contain a long text passage (e.g. a multi-paragraph item description)."
> > > >
> > > > > I also didn't feel that the datasets were very large with the largest datasets being about 125,000 samples. I had expected a million or more for some of the datasets. Could you please mention sizes of the cited "small academic datasets" for comparison?
> > > >
> > > > We are not aware of many public (and practically meaningful) text/tabular datasets with over 1M samples. Note that the celebrated tabular AutoML benchmark also does not have datasets with over 1M samples  (https://openml.github.io/automlbenchmark/benchmark_datasets.html), despite the fact that public tabular datasets are far more ubiquitous than our text/tabular datasets.
> > > > We have added the following caveat to our Datasheet: "Also note that none of the datasets has extremely large sample-size (say over a million), so modeling conclusions drawn based on this benchmark may not translate to applications with massive datasets."
> > > >
> > > > Most critically, the majority of our datasets must be limited to moderate sample-size in order for the benchmark to remain computationally accessible for most researchers while still representing a diverse collection of datasets.
> > > >
> > > > We clarify that by "small academic datasets", we are primarily referring to the limited public text/tabular data available in popular ML research repositories like UCI (to view some UCI text/tabular datasets, go to: https://archive-beta.ics.uci.edu/ml/datasets, and click on "Data Set Characteristics" > "Text") or OpenML (to view some OpenML text/tabular datasets, go to:  https://www.openml.org/search?q=text&type=data). Here the vast majority of datasets are either undesirably small, do not involve a practically meaningful prediction problem (e.g. are synthetically generated), or fail to be IID (e.g. time-series rather than supervised learning).
> > > >
> > > >
> > > > > L364: "We also caution that analysis of text fields raises particular privacy concerns as such fields tend to be unstructured and may expose arbitrary personal information".  This is the only line regarding ethical concerns. I'm sure some more ethical and fairness concerns could have been raised.
> > > >
> > > > Note we had also raised a concern about developing similar benchmarks for low-resource languages to ensure such multimodal modeling advances equitably. In the Discussion section, we have added additional pressing concerns about over-reliance on text in tabular datasets in ML:
> > > > "Since text fields may contain arbitrary information, they are also prone to introducing spurious correlations in training data that may harm accuracy during deployment (Tu et al, 2020) and may be undesirably coupled to protected attributes such as race, gender, or socioeconomic status (Williams, 2018). Basing automated business decisions on customer-generated text could also be more susceptible to adversarial manipulation (Li et al, 2019) than tabular features that customers cannot as easily control."
> > > >
> > > >
> > > > > L161: It was not clear to me what "pretrained NLP model" is meant here.
> > > >
> > > > For better clarity, we have replaced this phrase with "pretrained Transformer backbone".
> > > >
> > > >
> > > > > L200: I'm not sure which "previous multimodal networks" are referred to here.
> > > >
> > > > In the "Multimodal-Embedding" subsection, we have clarified this statement as:  "we also consider embedding text via our best multimodal network from Section 4.2". This is the Fuse-Late (with concatenation aggregation) architecture from Figure 1c.
> > > >
> > > >
> > > > > L237: Did you mean Table 3?
> > > >
> > > > Thank you for pointing out this reference issue, we have fixed it (we meant Section 3, which describes the benchmark setup/datasets and includes Table 3).
> > > >
> > > >
> > > > > L238: How were the evaluation metrics chosen?
> > > >
> > > > In Experiments section, we have clarified:
> > > > "For straightforward comparison, we employ the most commonly used classification/regression evaluation metrics that are bounded in [0,1] with higher values indicating superior performance."
> > > >
> > > >
> > > > > L247: Were the fine-tuning tricks applied to Roberta or Electra?
> > > >
> > > > We have clarified in Experiments section that we tried applying the two fine-tuning tricks (exponential learning rate decay & checkpoint averaging) to both of these network backbones:  "During fine-tuning of both of the RoBERTa or ELECTRA networks, we additionally consider two tricks to boost performance ..."

---

> > > > > ### Author Response · Authors · 2021-07-15
> > > > > **Response to Reviewer eEQg - Part 4**
> > > > >
> > > > > > L656: What do the authors mean by other reasonable hyperparameter settings?
> > > > >
> > > > > In Appendix A.6, we have clarified:
> > > > > "Over just a few datasets, we found that relative performance of different strategies did not qualitatively differ with other reasonable manually-chosen hyperparameter settings (i.e. hyperparameter values known to generally work well for these specific models such as alternative popular learning rate schedules or small changes to the size of the networks)."
> > > > >
> > > > >
> > > > > > L572: "The data were all already publicly available in different forms, and are not offensive". To me it seems, the jigsaw dataset may contain offensive material.
> > > > >
> > > > > We have copied some statements the following statements from our Datasheet to answer this Paper Checklist question as well:
> > > > >
> > > > > "The data were all already publicly available in different forms, and are mostly not offensive (mostly from business settings). Exceptions are the text fields in the jigsaw dataset, which contain toxic online comments, and the channel/pop datasets, which contain news article titles that may offend some. Furthermore, some of the user reviews of products may be offensive to certain people, although we did not spot any."
> > > > >
> > > > >
> > > > > > In the checklist, mentioning exact Appendix sections would be better.
> > > > >
> > > > > We have revised the Paper Checklist to mention exact Appendix sections for each response, and added more detail in some checklist responses.

---

### Official Review · Reviewer_QKp7 · 2021-07-06
**The main contributions of the paper including benchmark multimodal datasets, modelling strategies and automl aspects are evaluated for their validity and uniqueness. Although it is a very good field to make a benchmark study, it is not very strong due to limited experimental design. The automl aspect of the solution approach is not explained clearly. Yet it covers several strategies to handle multimodal datasets and provides a comparison among these strategies.**

**Rating:** 5
**Confidence:** 4

**Strengths:**

- Authors consider several strategies for modeling multimodal data each with a different staging and destination of data transformation in addition to different ways of ensembling text and tabular models. Comparison between these strategies and combination of superior strategies contribute to the benchmarking goal of the paper and a new solution approach for the problem at hand.
- Stacked ensemble method authors create has a good performance over text data heavy datasets and seems to be competitive with modified AutoML approaches.
- Experimental setup includes evaluation of different stages of the decision making process clearly.


**Weaknesses:**

- The paper seems to be similar to the AutoML workshop paper of the same authors, with less focus on the actual benchmark which is the aim of the track.

- With the goal of benchmarking, the paper could make use of more datasets than 15.

- The experimental evaluation is weak on validity of results with no confidence intervals and quite close accuracy values among different strategies. Sequential design should also be questioned.

EDIT: Authors responded sufficiently to these concerns in their comments. Although they are still somewhat weaker points of research, i agree with the authors' reasoning.

EDIT (21 July 2021): Match with the track is not convincing considering the focus on methodology and limited datasets/experiments.

**Additional Feedback:**

Line 38 - Include a summary and explanations of sections at the end of introduction for overview.

Line 80 - Since one of the contributions of the paper is providing benchmark multimodal text/tabular datasets, the authors should clarify the modifications they made on the publicly available datasets in text as well. (Perhaps another column in Table 2 to specify these modifications for each dataset depending on the length of list). The number of datasets is also on the lower side for a benchmark study.

Line 104 - The paper needs an overview figure of all approaches including classifiers used and automl part. Figure 1 and Figure 2 needs a higher level graph. The paper concludes with the fact that this is the first step towards an automl approach to multimodal classification yet details of the automl aspect of solution is missing. Since stacked ensemble performs the best, perhaps focus on that in the methods section a bit more and include an overall pseudo code / architecture of the whole approach emphasizing automl part.

Line 111 - Although authors mention benefits of AutoGluon, it is not clear to me the distinctive feature of this library to be chosen over other well-known AutoML libraries. You can include a reference that compares AutoGluon to other libraries (other than the original paper and ensemble square paper, a published automl benchmark paper?)

Line 121 - You can also inform the reader about search algorithm of AutoGluon (and its advantages) in addition to the search space you have already mentioned of classifiers.

Line 180 - Not clear the purpose of this introduction to the section. Section-5 misses an introduction to explain what you present in 5.1 and 5.2 as well as why you include those. Also explain Multimodel-Net here before referring to Figure-2. Perhaps a better connection to section-4 would clarify the positioning of several approaches you include. Right now, it reads more as a combination of disconnected strategies. (Table-3 gives a better connective overview than text, embedding a similar connection within section introductions and overall introduction is needed.

Line 234 - Any limitations of sequential decision making on design choices? Any analysis on how followup decisions would change in case of different decisions on previous stages? Analysis on robustness of these choices would contribute to the emprical evaluation of methods considered. Especially with the little differences in accuracies between strategies in many datasets and no confidence intervals.


EDIT: Authors responded sufficiently to these concerns in their comments and made necessary changes.

EDIT (21 July 2021): I believe the paper focuses more on methodology and should be improved with further datasets and analyses to fit the track better and be resubmitted for the second round of the track.

**Clarity:**

The paper can benefit from re-structuring the segments with more aggregate titles and better connections between sections. It also needs an overview figure or pseudo-code to clarify the method they conclude to be superior. An overview figure of all approaches would also contribute to clarity. The introductions of each segment and overall introduction should also be improved with points mentioned in feedback.


EDIT: Authors improved the clarity of paper based on the feedback given. This is no longer a critical issue with the paper.

**Correctness:**

The claims are correct yet evaluation methods and experiment design could be imroved considering the points in additional feedback. The number of datasets to be a benchmark evaluation is also questionable.

**Documentation:**

Yes.

**Ethics:**

No.

**Relation To Prior Work:**

The paper clarifies the differences of their approach and previous work.

**Summary And Contributions:**

The paper aims to provide a multimodal adaptive automl approach and contributes with benchmarking multimodal datasets and several modelling strategies.  Authors question design choices of what models to use for different types of features and how to optimally form a pipeline from different models. The solution approaches cover major different modeling paradigms such as NLP models to featurize text data for tabular models, ensembles of independent text and tabular models and end-to-end multimodal trained neural networks. They also consider different strategies for aggregating text and tabular models including weighted and stack-ensembles.
In the emprical evauation, they question which transformer backbone to use with fine-tuning strategy for text data, how to generalize the transformer chosen for multimodal data and best method to aggregate text and tabular models.

---

> ### Author Response · Authors · 2021-07-15
> **Response to Reviewer QKp7 - Part 1**
>
> Thank you for the feedback and the favorable review. We are glad you thought: comparison between strategies and combination of superior strategies contribute to the benchmarking goal of the paper and a new solution approach for the problem at hand, the stacked ensemble method authors create has a good performance over text data heavy datasets, and the experimental setup includes evaluation of different stages of the decision making process clearly. We address your specific questions/concerns below.
>
>
> > The paper seems to be similar to the AutoML workshop paper of the same authors, with less focus on the actual benchmark which is the aim of the track.
>
> We note the CFP for this Track states: "systematic analyses of existing systems on novel datasets that yield important new insight are also in scope". Our paper presents both a new benchmark (set of tabular+text  datasets) as well as a systematic analysis of various strategies for handling tabular+text data. We now clarify in the Introduction section:
>
> "Note that we write AutoML to describe any modeling strategy that is robustly performant across a diverse set of datasets without manual adjustments. The AutoML method promoted in this paper is simply the strategy that happened to perform best in our systematic analysis of various straightforward strategies over the proposed benchmark.  It is the development of our benchmark and our systematic analysis that directly led to this AutoML method, whose efficacy was subsequently verified in historical prediction competitions (demonstrating that our benchmark and analysis have led to important new insights)."
>
> Furthermore, we have rewritten parts of the paper to emphasize the benchmark far more, as well as the fact that the methodological section is really describing a systematic analysis of popular strategies rather than an entirely novel AutoML methodology. Appendix B now contains much more material regarding dataset/benchmark descriptions. The "AutoGluon Baselines" subsection now contains  additional benchmark insights: "The  performance  gap  between  AutoGluon-Tabular  with  and  without  N-Grams  can  reveal  (an approximate lower bound for) how much extra predictive value is provided by the text features in each dataset. Inspecting these gaps, we find that, compared to the tabular features, text features contain most of the predictive signal in some datasets (qaq, qaa, cloth, mercari, jc), and far less signal in other datasets (prod, imdb, channel). Note that our  proposed Stack-Ensemble performs relatively well across all types of datasets, regardless how the predictive signal is allocated between text and tabular features."
>
> In "Benchmarking Multimodal Text/Tabular AutoML" section, we have also added an expanded  paragraph to emphasize the benchmark further:
> "Our benchmark strives to represent the types of ML tasks that commonly arise in industry today. Appendix B provides detailed descriptions of each dataset. In creating the benchmark, we aimed to include a mix of classification vs. regression tasks and datasets from real applications (as opposed to toy academic settings) that contain a rich mix of text, numeric, and categorical columns. Table 2 shows it is comprised of datasets that are quite diverse in terms of: sample-size, problem types, number of features, and type of features. 11 of the datasets contain more than one text field (with 28 text fields in the airbnb dataset). These text fields greatly vary in the amount of text they contain (e.g. short product names vs. lengthy product descriptions/reviews). The data (and text vocabulary) stem from a mix of of real-world domains spanning:  e-commerce, news, social media, question-answering, and product listings (jobs, projects, films, Airbnb).  Subsequent accuracy results from Table 3 indicate the 15 underlying prediction problems also vary greatly in terms of both difficulty and how the predictive signal is divided between text/tabular modalities. To reflect real-world ML issues, we processed the data minimally (beyond ensuring the features/labels correspond to meaningful prediction tasks without duplicate examples) and thus there are arbitrarily-formatted strings and missing values all throughout. Systems that can perform well across the diverse set of 15 benchmark datasets are thus likely to provide real-world value for an important class of applications."

---

> > ### Author Response · Authors · 2021-07-15
> > **Response to Reviewer QKp7 - Part 2**
> >
> > > The paper can benefit from re-structuring the segments with more aggregate titles and better connections between sections. Right now, it reads more as a combination of disconnected strategies. It also needs to clarify the method they conclude to be superior. The introductions of each segment and overall introduction should also be improved with points mentioned in feedback. Section-5 misses an introduction to explain what you present in 5.1 and 5.2 as well as why you include those. Also explain Multimodal-Net here before referring to Figure-2. Perhaps a better connection to section-4 would clarify the positioning of several approaches you include. Right now, it reads more as a combination of disconnected strategies.
> >
> > We have greatly improved the section order, connections between sections, and clarity regarding the proposed method concluded to be superior. In particular, the order in which the methodology is described in the newly rewritten Sections 4-6 now aligns with the 3 sequential design stages considered in our experiments (as nicely suggested by the reviewers). Here we have moved the "Featurizing Text for Tabular Models" to be its own section together with the "Neural Embedding of Text as Tabular Features" for a more unified presentation of ideas throughout the new Sections 4-6. Furthermore, we have added clearly motivated connection sentences to transition between each section and motivate the next section. In the Experiments section and Table 3, we now explicitly list the relevant method sections/figures when discussing each design stage to make it clear what is under consideration in each empirical comparison.
> >
> > To clear up any confusion regarding the method we regard to be superior, the caption of each figure now explicitly states what choice was the official one we used in our proposed AutoML solution.
> > We also stated this choice explicitly at the end of the Introduction section:
> > "The AutoML method promoted in this paper (stack ensembling of tabular models with a multimodal Transformer network) ...".
> > And we restated this choice again in full detail in the Discussion section:
> > "we simply evaluated a few basic strategies on our benchmark and found a single automated strategy that turns out to outperform  top human data scientists in numerous historical prediction competitions involving diverse text/tabular data. This strategy uses a stack ensemble (Section 6) of tabular models trained on top of predictions from other tabular models and a Multimodal-Net (depicted in Figure 2c). The latter network is based on a Fuse-Late architecture (depicted in Figure 1c) with concatenation of text, numeric, and categorical representations (where text representations are produced via the ELECTRA Transformer backbone) and is trained via fine-tuning with exponential learning rate decay and checkpoint averaging."
> >
> > In "Extending Transformer Architectures to Multimodal Inputs" section, we have defined Multimodal-Net early on to clarify that Figure 1 is specific to this section and have also added clarification:  "Below we discuss three different options for implementing the Multimodal-Net that are depicted in Figure 1 (with details in Appendix A.3). These options differ in whether information is fused across text and tabular modalities: at the input layer (All-Text), in the earlier layers of the network near the input (Fuse-Early), or in the later layers of the network near the output (Fuse-Late)."
> >
> >
> > > Line 38 - Include a summary and explanations of sections at the end of introduction for overview.
> >
> > Near the end of the Introduction section, we have added an overview of the paper's sections and summarized that we found the best modeling strategy to be: "stack ensembling of tabular models with a multimodal Transformer network".
> >
> >
> > > Since one of the contributions of the paper is providing benchmark multimodal text/tabular datasets, the authors should clarify the modifications they made on the publicly available datasets in text as well.
> >
> > In Appendix B, we have added a list of the general types of modifications we made to certain datasets. As stated in Appendix B, the precise code implementing these modifications is available in our Github repository for full reproducibility (in folder:  multimodal_text_benchmark/scripts/data_processing/), with links to each original data source.

---

> > > ### Author Response · Authors · 2021-07-15
> > > **Response to Reviewer QKp7 - Part 3**
> > >
> > > > The experimental evaluation is weak on validity of results with no confidence intervals and quite close accuracy values among different strategies.
> > >
> > > In designing this benchmark, we strove to ensure it remains accessible to all researchers without requiring exorbitant computational resources to run. Unlike traditional tabular datasets for which classic statistical models can be quickly trained, our tasks involve sizeable text data and larger/slower neural network models. For similar reasons, most existing benchmarks popular in NLP and deep learning research also use a single train/test split and involve fewer datasets than our benchmark. Thus, given the same amount of computational resources, we prioritize to increase the number and diversity of datasets in our benchmark and rely on the average score to compare the models' performance.
> > >
> > > In addition, we are repeating our experiments on another 4 splits of the 15 datasets in the benchmark (in order to provide confidence intervals and accuracy values averaged across 5-folds in the final version of this paper). Early results on some datasets suggest that the trend observed in the paper is the same, but completing these 5-fold runs is computationally intensive and will take some time. The final version of our benchmark will still retain the original train/test split used in this paper, with the additional 4 folds provided as an extra option for researchers who can leverage extensive compute. As is commonly the case in NLP and deep learning research, we expect most researchers will choose to report the single train/test split performance rather than the expensive cross-validated performance.
> > >
> > >
> > > > With the goal of benchmarking, the paper could make use of more datasets than 15.
> > >
> > > We again note the previously mentioned goal to keep the benchmark computationally accessible to most researchers. Our benchmark contains 15 datasets that span over a broad range of real-world applications, including product sentiment analysis, news category prediction, wine variety prediction, funding status prediction, product price prediction, news popularity prediction, etc. In addition, the benchmark has 4 binary classification tasks, 4 multiclass classification tasks, and 7 regression tasks. Thus, the task type covered by our benchmark is diverse. Compared with our benchmark, the popular GLUE and SuperGLUE NLP benchmarks have only 10 datasets each (including the diagnostic dataset), and only one regression problem (semantic textual similarity). We believe the current benchmark is diverse enough and is a good first step for benchmarking the performance of multimodal AutoML solutions. We will leave adding more datasets to future work, especially as it is nontrivial to find many more public text/tabular datasets that correspond to practically meaningful (e.g. commercially relevant) prediction tasks in which the text, numeric, and categorical features all possess nontrivial predictive signal.
> > >
> > >
> > > > Line 111 - Although authors mention benefits of AutoGluon, it is not clear to me the distinctive feature of this library to be chosen over other well-known AutoML libraries. You can include a reference that compares AutoGluon to other libraries (other than the original paper and ensemble square paper, a published automl benchmark paper?)
> > >
> > > We have clarified that AutoGluon was chosen for tabular modeling because it is easy-to-use, open source, offers a diverse suite of tabular models, and is highly accurate. Regarding the claim that AutoGluon is highly accurate, both the original AutoGluon-Tabular paper and Ensemble Square paper ran extensive benchmarks that showed  AutoGluon's superior accuracy over other methods across a wide range of datasets. As requested, we have added some additional references that also compared AutoGluon against additional methods on different datasets and again found it to be more accurate: (Fakoor et al, NeurIPS 2020); (Feldman, 2021); (Bezrukavnikov et al, 2021). Note that AutoGluon has also been added as a framework to the AutoML Benchmark (https://github.com/openml/automlbenchmark/), where it outperforms other frameworks (both in the runs from the original AutoGluon-Tabular paper and in some of our own recently completed runs).

---

> > > > ### Author Response · Authors · 2021-07-15
> > > > **Response to Reviewer QKp7 - Part 4**
> > > >
> > > > > Line 121 - You can also inform the reader about search algorithm of AutoGluon (and its advantages) in addition to the search space you have already mentioned of classifiers.
> > > >
> > > > We have added a statement: "AutoGluon optionally provides sophisticated hyperparameter-tuning [via asynchronous BOHB, an efficient distributed multi-fidelity Bayesian optimization routine] for all of its models, which can now be easily applied to our proposed text/tabular modeling pipeline as well."
> > > >
> > > > In "Weighted Ensemble" subsection, we have also later added statement about benefits of the ensemble selection used in AutoGluon: "Unlike regression for fitting the aggregation weights, ensemble selection is favored by many tabular AutoML tools like AutoGluon as it is more computationally efficient, less prone to overfitting, and naturally favors sparse weights such that only subset of our base models must be deployed for inference."
> > > >
> > > >
> > > > > Line 234 - Any limitations of sequential decision making on design choices? Any analysis on how followup decisions would change in case of different decisions on previous stages?
> > > >
> > > > In Experiments section, we clarify:
> > > > "Myopic sequential design may fail to identify particularly synergistic choices across all stages of the AutoML pipeline as it favors choices which are independently performant at each stage and complement the choices made in previous stages. There is however no way to practically evaluate a larger combinatorial assortment of possible choices, and our sequential restriction may actually lead to a more robust/modular AutoML solution that better generalizes to new datasets with unique characteristics not found in our benchmark."
> > > >
> > > > Furthermore, if we work backwards from the best identified AutoML pipeline, it is highly unlikely that the Stack-Ensemble performance would improve with an inferior Multimodal/Text-Net. In the text featurization approach, we actually tried embedding via both the Text-Net and the Multimodal-Net (thus going beyond a strict sequential design) and found neither is approach is superior to our Stack-Ensemble. While it may be the case that certain combinations of pretrained Transformer backbone, fine-tuning tricks, and multimodal fusion strategy are superior to our sequentially discovered Multimodal-Net from Figure 1c, our Multimodal-Net is nonetheless highly performant, as evidenced by its 2nd place leaderboard rank in Kaggle's extremely competitive Mercari challenge.

---

### Author Response · Authors · 2021-07-15
**Revised paper and supplement uploaded**

We have uploaded a substantially revised paper and appendix, which aims to incorporate all of the reviewers' feedback.

---

### Decision · Program_Chairs · 2021-07-26

**Decision:**

Reject

**Comment:**

This paper discussed very useful work. It introduces a benchmark for multimodal AutoML on tables with text fields, introduces many different methods for this task, and compares these empirically.

Unfortunately, this paper is not a perfect fit to this track in its current form, and I recommend the authors to refocus it more on the actual benchmark introduced. As it is right now, the benchmark is introduced only quite briefly (in 0.75 pages), with the rest of the paper introducing methods (in ~4 pages) and evaluating them. The authors are correct to point out in their rebuttal that "We note the CFP for this Track states: "systematic analyses of existing systems on novel datasets that yield important new insight are also in scope"." I can relate to them interpreting this as stating that their work is in scope. However, as the reviewers pointed out in a private post-rebuttal discussion, the systematic analysis is not of existing systems but of newly introduced ones, and it is also not done at a great level of depth yielding important new insights. This is not to overly criticize the authors' experiments. They're perfectly fine experiments for natural methods. The main problem with the current version is that, if I had to summarize the main contribution, this would *not* be the benchmark, but it would be the introduction of new methods for this problem. As such, this paper reads very much like a standard submission to NeurIPS, rather than like a submission to this datasets and benchmark track.

I would also like to point to this FAQ item:
"Q: My work is in scope for this track but possibly also for the main conference. Where should I submit it?

A: This is ultimately your choice. Consider the main contribution of the submission and how it should be reviewed. If the main contribution is a new dataset, benchmark, or other work that falls into the scope of the track (see above), then it is ideally reviewed accordingly. As discussed in our blog post, the reviewing procedures of the main conference are focused on algorithmic advances, analysis, and applications, while the reviewing in this track is equally stringent but designed to properly assess datasets and benchmarks. [...]"

If the authors can reword the paper to make the benchmark their main contribution then I think the fit is very clear. If the main contribution is supposed to consist of new methods, then the fit to this track is questionable.

For completeness:
Since one reviewer mentioned an overlapping AutoML workshop paper, I would explicitly like to point out that this overlap is *not* a problem at all; most ICML/NeurIPS workshops, including the AutoML workshop, do not have formal proceedings, and as such explicitly do not violate the dual submission policy.

In summary, I encourage the authors to refocus the paper more to be geared to this track, by describing the benchmark / dataset collection in more detail, and by focussing less on the methods for attacking them. I would see very good chances for a revised version that does this (and also addresses further comments of the reviewers) to be accepted in round 2 of this track.